# A metagenomic DNA sequencing assay that is robust against environmental DNA contamination

Omary Mzava[1,11], Alexandre Pellan Cheng[1,11], Adrienne Chang [1,11], Sami Smalling[1],
Liz-Audrey Kounatse Djomnang [1], Joan Sesing Lenz[1], Randy Longman[2], Amy Steadman[3],
Luis G. Gómez-Escobar[4], Edward J. Schenck [4], Mirella Salvatore [5], Michael J. Satlin[6],
Manikkam Suthanthiran[7,8], John R. Lee [7], Christopher E. Mason [9,10], Darshana Dadhania [7,8] &
Iwijn De Vlaminck [1]✉

Metagenomic DNA sequencing is a powerful tool to characterize microbial communities but is sensitive to environmental DNA contamination, in particular when applied to samples with low microbial biomass. Here, we present Sample-Intrinsic microbial DNA Found by Tagging and sequencing (SIFT-seq) a metagenomic sequencing assay that is robust against environmental DNA contamination introduced during sample preparation. The core idea of SIFT-seq is to tag the DNA in the sample prior to DNA isolation and library preparation with a label that can be recorded by DNA sequencing. Any contaminating DNA that is introduced in the sample after tagging can then be bioinformatically identified and removed. We applied SIFT-seq to screen for infections from microorganisms with low burden in blood and urine, to identify COVID-19 co-infection, to characterize the urinary microbiome, and to identify microbial DNA signatures of sepsis and inflammatory bowel disease in blood.

[1] Nancy E. and Peter C. Meinig School of Biomedical Engineering, Cornell University, Ithaca, NY, USA. [2] Division of Gastroenterology and Hepatology, Weill Cornell Medicine, Jill Roberts Center for IBD, New York, NY, USA. [3] Global Health Labs, Bellevue, WA, USA. [4] Division of Pulmonary and Critical Care Medicine, Department of Medicine, Weill Cornell Medicine, New York, NY, USA. [5] Divisionof Public Health Programs, Department of Medicine, Weill Cornell Medicine, New York, NY, USA. [6] Division of Infectious Diseases, Department of Medicine, Weill Cornell Medicine, New York, NY, USA. [7] Division of Nephrology and Hypertension, Department of Medicine, Weill Cornell Medicine, New York, NY 10065, USA. [8] Department of Transplantation Medicine, New York Presbyterian Hospital–Weill Cornell Medical Center, New York, NY 10065, USA. [9] Department of Physiology and Biophysics, Weill Cornell Medical College, New York City, NY, USA. [10] WorldQuant Initiative for Quantitative Prediction, New York, NY 11238, USA. [11]These authors contributed equally: Omary Mzava, Alexandre Pellan Cheng, Adrienne Chang. ✉email: vlaminck@cornell.edu

Metagenomic DNA sequencing is a routinely used tool to characterize the genetic makeup and species composition of microbial communities. In addition, metagenomic DNA sequencing of clinical samples is increasingly used for unbiased detection of microbial infection. Nonetheless, sample contamination by environmental DNA plagues these assays. DNA contamination unavoidably occurs to a degree during the process of sample preparation for DNA sequencing and is particularly problematic for samples that have a low biomass of microbial DNA that can easily be overwhelmed by contaminating DNA[1–3].

Multiple solutions have been proposed to overcome the impact of DNA contamination on low biomass metagenomic sequencing. DNA contamination can be avoided to an extent by processing samples in a clean room facility, sterilizing consumables, and incorporating non-redundant dual indexing and unique molecular identifiers during library preparation[4,5]. However, while these approaches minimize the influence of contaminant DNA, they do not avoid contaminant DNA present in reagents. Other approaches are based on batch-correction algorithms that identify microbial species detected in negative controls, those in low relative abundance, or those that are inversely correlated with DNA concentration[5,6]. These indirect methods of identifying contaminant species, however, tend to overcorrect, eliminate sample-intrinsic species that are also common DNA contaminants, and make the incorrect assumption that sample contamination is perfectly reproducible across all samples in a batch. Here, we describe Sample-Intrinsic microbial DNA Found by Tagging and sequencing (SIFT-seq), a metagenomic sequencing method that is robust against DNA contamination. SIFT-seq tags sample-intrinsic DNA directly in plasma and urine with a chemical label that can be recorded via DNA sequencing. Contaminating DNA that is introduced in the sample after this initial tagging step can then be directly identified and eliminated. Several biochemistries can be envisioned for the initial DNA tagging step. Here, we implement deamination of unmethylated cytosines via bisulfite salt treatment of DNA. This chemistry does not require the use of enzymes or DNA oligos and can be applied directly to clinically-relevant samples, such as blood and urine, as demonstrated in this work. We present an analysis of the technical performance of SIFT-seq and describe six proof-of-principle applications of SIFT-seq: (i) to identify viral and bacterial COVID-19 co-infection from blood; (ii) to screen for urinary tract infection (UTI); (iii) to characterize the urinary microbiome; (iv) to screen for infections with low burden and prevalence in the blood of patients that presented with respiratory symptoms at outpatient clinics in Uganda; (v) to identify microbial DNA signatures in the blood of patients with sepsis and (vi) inflammatory bowel disease (IBD).

## Results

**SIFT-seq working principle.** For the practical implementation of SIFT-seq, we tag DNA by bisulfite salt-induced conversion of unmethylated cytosines to uracils (Fig. 1a). Uracils created by bisulfite treatment are converted to thymines in subsequent DNA synthesis steps that are part of DNA sequencing library preparation. After DNA sequencing, contaminating DNA introduced after tagging can then be directly identified based on the lack of cytosine conversion. Bisulfite conversion does not require the use of commercial enzymes or oligos that are a frequent source of DNA contamination, and we found that it can be applied directly to the original sample, before DNA isolation. We developed a bioinformatics procedure to differentiate sample-intrinsic microbial DNA, contaminant microbial DNA, and host-specific DNA after SIFT-seq tagging (Fig. 1b, Methods). This procedure

consists of three steps. First, host cfDNA is removed via mapping and k-mer matching. Given that CpG dinucleotides are heavily methylated in the human genome and rarely in microbial genomes, sequences containing CG dinucleotides are also removed. Second, remaining sequences that consist of more than three cytosines, or one cytosine-guanine dinucleotide are flagged and removed as likely contaminants. Last, a species-level filtering step is performed to remove any remaining reads that primarily originate from C-poor regions in the reference genome (Fig. 1c, Methods).

We devised two assays to test the principle of SIFT-seq. First, we applied SIFT-seq and conventional DNA sequencing to samples of sheared ΦX174 DNA (New England Biolabs, #N3021S) with variable biomass (0.0025 ng, 0.025 ng, 0.25 ng, 2.5 ng, 26 ng, and 155 ng for SIFT-seq; 0.004 ng, 0.04 ng, 0.4 ng, 4 ng, 35 ng, and 240 ng for standard cfDNA sequencing). We first quantified the abundance of *Cutibacterium acnes (C. acnes)*, which is a frequent member of the normal skin flora and is routinely identified as a contaminant in DNA sequencing[7]. We observed an increase in *C. acnes* abundance with decreasing input biomass, as expected given that samples with a lower biomass are more susceptible to environmental contamination (Fig. 1d). We found that despite a ~30% lower biomass at the beginning of library preparation for the SIFT-seq samples, far fewer *C. acnes* reads were present after SIFT-seq filtering (4223.8 and 119.5 MPM in the highest biomass samples, 1.48 and 0 MPM in the lowest biomass samples, before and after SIFT-seq filtering respectively; Fig. 1d).

Second, we performed SIFT-seq on sheared ΦX174 DNA samples with variable biomass (0.0025–155 ng; Fig. 1e) which we spiked after SIFT-seq tagging with 1 ng of sheared DNA from a well-characterized community of microbes to simulate microbial DNA contamination (10 species; Zymo Research, #D6305). Before applying the SIFT-seq bioinformatics filter, we observed a negative correlation between the ΦX174 DNA input biomass and the relative number of reads from the spike-in community, as expected (Pearson's $R = -0.54$, $p$ value $= 6.5 \times 10^{-6}$; Spearman's $\rho = -0.82$, $p$ value $= 6.3 \times 10^{-16}$; Fig. 1e). After applying the SIFT-seq filter, we observed an average percent decrease of 99.8% of molecules mapping to species of the spike-in community (Fig. 1f). Sequences mapping to *Escherichia coli (E. coli)* were the most abundant after filtering (58.89%). Given that ΦX174 genomic DNA is isolated after phage propagation in *E. coli* culture, we reasoned that these remaining reads were likely intrinsic to the original sample. Together, these experiments demonstrate the effectiveness of SIFT-seq for the detection and removal of DNA contaminants without removing species originally present in the sample.

**Application of SIFT-seq to cell-free DNA in blood and urine.** Cell-free DNA (cfDNA) in blood and urine has emerged as a useful analyte for the diagnosis of infection[8–15]. Metagenomic cfDNA sequencing can identify a broad range of potential pathogens with high sensitivity. Yet, because of the low biomass of microbial-derived cfDNA in blood and urine, metagenomic cfDNA sequencing is highly influenced by environmental contamination, limiting the specificity of metagenomic cfDNA sequencing for pathogen identification.

To assess the performance of SIFT-seq in metagenomic cfDNA sequencing, we assayed a total of 196 cfDNA samples (154 plasma, 42 urine) collected from five groups of subjects: (1) 30 plasma samples from a cohort of 14 patients hospitalized with COVID-19 ("COVID19 cohort"), (2) 53 plasma samples from a cohort of 44 patients seeking treatment for IBD (4 patients without IBD, 19 patients with Crohn's disease, 21 patients with ulcerative colitis; "IBD cohort"), (3) 56 plasma samples from a cohort of 44 patients presenting with respiratory symptoms at

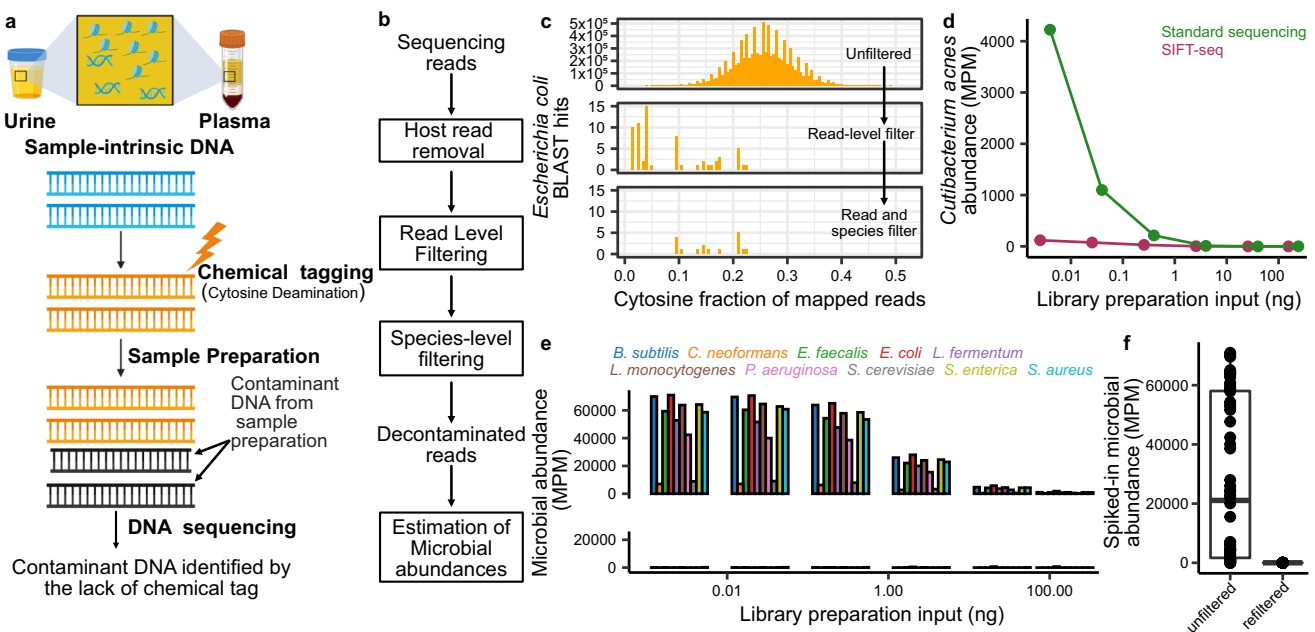

**Fig. 1 SIFT-seq proof-of-principle. a** Experimental workflow. Tagging of sample-intrinsic DNA by bisulfite DNA treatment is performed directly on urine or plasma. Contaminating DNA introduced after the tagging step is identified based on lack of cytosine conversion. **b** Bioinformatics workflow. **c** Representative example of the cytosine fraction of mapped reads in an unfiltered (top) dataset, a read-level filtered dataset (middle) and a fully filtered dataset (bottom). **d** Number of reads assigned to *Cutibacterium acnes* (common environmental DNA contaminant) in ΦX174 DNA after conventional sequencing (green) and SIFT-seq (purple). **e** Deliberate contamination assay. Detection of known contaminants before (top) and after (bottom) filtering. **f** Number of reads assigned to contaminants. Boxes in the boxplots indicates 25th and 75th percentile, the band in the box indicated the median and whiskers extend to 1.5 × Interquartile Range (IQR) of the hinge. Outliers (beyond 1.5 × IQR) are plotted individually. Source data for (**d–f**) are provided as a Source Data file.

outpatient clinics in Uganda ("Uganda cohort"), (4) 15 plasma samples from a cohort of 15 patients (10 patients with sepsis, 5 patients without sepsis but in the ICU; "sepsis cohort"), (5) 26 urine samples from a cohort of kidney transplant patients with and without urine culture-confirmed UTIs (16 positive urine culture, 10 negative urine culture; "kidney transplant cohort") and (6) 16 urine samples collected early after transplantation from 10 kidney transplant patients that received a ureteral stent at the time of transplantation (samples were collected pre-stent and post-stent removal for 5 of the 10 patients; "early post-transplant cohort"; see Supplementary Table 1 and Supplementary Information for details on the patients and samples included).

We performed SIFT-seq for all samples and obtained an average of 48.5 ± 23.4 million paired-end reads per sample. We detected and quantified the abundance of 68 genera that have been reported as frequent DNA contaminants in multiple independent studies (summarized in Ref. [4]; Fig. 2a; 49 of these genera detected in at least one sample). We found that 77% of these genera were completely removed from all samples after SIFT-seq filtering. We calculated the total number of molecules from all contaminant genera and observed an up to three orders of magnitude reduction after SIFT-seq filtering (reduced by a factor of 7.5, 1,711.2, 177.6, 608.8, 215.4, 547.2; two tailed, Wilcoxon signed-rank test, *p* values < 0.001 for all cohorts; Fig. 2b). We investigated the impact of SIFT-seq filtering on removing reads originating from the skin contaminant *C. acnes* (Fig. 2c). *C. acnes* was detected in all samples and completely removed from 62 samples by SIFT-seq filtering. In the remaining samples, we observed an up to two orders of magnitude reduction of *C. acnes* reads (two tailed, Wilcoxon signed-rank test, *p* values < 0.001 for all cohorts).

We next evaluated the utility of SIFT-seq to correct for batch effects and to reveal true differences in microbiome profiles for

different patient groups. To this end, we calculated the Bray–Curtis Dissimilarity Index for all clinical samples included in this study and sorted the datasets based on the following parameters: (1) sequencing run, (2) operator, (3) urine culture test, (4) study cohort, and (5) biofluid type. Before SIFT-seq filtering, we observed a high similarity for samples assayed in the same experimental batches (Fig. 2d). SIFT-seq filtering removed these batch effects and revealed distinct cohort-specific microbiome profiles. Most notably, we observed distinct plasma microbiome profiles for plasma samples from the Uganda cohort (Fig. 2e). These results demonstrate that SIFT-seq directly applied to biofluids leads to a dramatic decrease in experimental noise and bias due to DNA contamination.

**SIFT-seq enables to screen for UTI and to characterize the urine microbiome.** The healthy urinary tract was long believed to be sterile[16,17], but this picture was challenged with recent advances in urine culture techniques that have identified bacteria in the urinary tract of both males and females[18]. Yet many microbes are difficult to cultivate in vitro, and bacterial culture can also be sensitive to contamination[19]. Therefore, comprehensive and accurate characterization of species colonizing the urinary microbiome is still lacking.

We reasoned that SIFT-seq could provide insight into the composition of the urine microbiome with both high sensitivity and specificity. We first applied SIFT-seq to 26 urine samples from 23 kidney transplant patients with and without infection of the urinary tract as determined by conventional urine culture (16 positive urine culture [*Enterococcus faecalis*: n = 3; *Enterococcus faecium*: n = 1; *E. coli*: n = 10; *Klebsiella pneumoniae*: n = 1; *Pseudomonas aeruginosa*: n = 1] and 10 negative urine culture). SIFT-seq consistently identified microbial cfDNA from species reported by urine culture

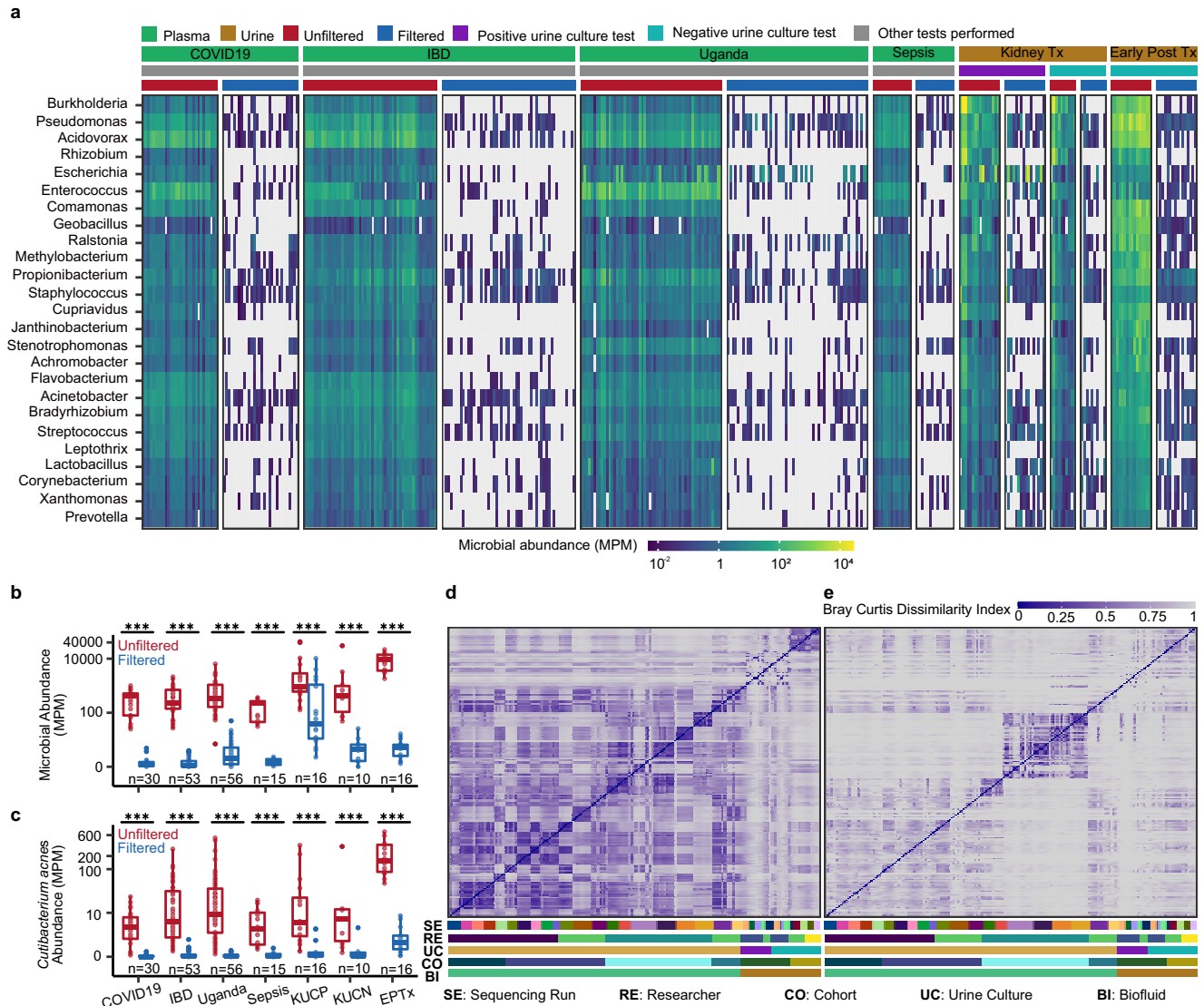

**Fig. 2 SIFT-seq applied to cell-free DNA in urine and plasma. a** Microbial abundance of 25 most abundant common contaminant genera (selected from the 68 genera[4]) before and after SIFT-seq filtering in plasma and urine from six independent subject cohorts (Tx = transplant). Total abundance of all contaminant genera (**b**) and *C. acnes* (**c**) before and after SIFT-seq filtering (KUCP = Kidney Transplant cohort with positive urine culture, KUCN = Kidney Transplant cohort with negative urine culture, EPTx = Early Post Transplant cohort). Bray–Curtis dissimilarity index before (**d**) and after (**e**) filtering. Samples are organized by: sequencing batch, researcher performing the experiment, cohort, and biofluid. Boxes in the boxplots indicates 25th and 75th percentile, the band in the box indicated the median and whiskers extend to 1.5 × Interquartile Range (IQR) of the hinge. Outliers (beyond 1.5 × IQR) are plotted individually. ***p value < 0.001. Source data are provided as a source data file.

(16/16 urine culture positive samples; Fig. 3a). SIFT-seq also identified two Corynebacterium species (*Corynebacterium jeikeium* and *Corynebaterium urealyticum*) in one sample from a urine culture positive patient (*E. coli*) with culture confirmed Corynebacterium co-infection. In addition, we found that samples from positive urine culture patients had a significantly higher burden of total microbial DNA compared to samples from negative urine culture patients (1451.8 ± 3024.7 MPM and 12.8 ± 17.6 MPM, respectively in the filtered samples; $p$ value = $7.1 \times 10^{-4}$, two tailed, Wilcoxon rank-sum test, Fig. 3b). Conventional metagenomic sequencing (without SIFT-seq filtering) detected uropathogens with equal sensitivity but was not robust against environmental contamination: DNA from common uropathogens not identified by culture was detected in many samples, albeit with low abundance, including in samples from patients without urine culture-confirmed UTIs. We conclude that the improved specificity of SIFT-seq allows for more accurate characterization of co-

infection networks in the scope of UTIs, and more accurate characterization of the normal urine microbiome in the absence of UTIs. It is important to note that two common skin microbes, *C. acnes* and *Staphylococcus epidermidis*, were found in most samples (23/26 samples). While these two species have been shown to cause UTIs[20,21], they may also have been introduced as contaminants at the time of urine collection, which underscores an important limitation of SIFT-seq: SIFT-seq is not robust against contamination that occurs before the tagging step.

Studies investigating the temporal dynamics of urine microbiome in individuals can benefit from the high sensitivity and specificity achieved with our assay. We applied SIFT-seq to paired urine samples obtained from five kidney transplant patients collected at two timepoints before and after ureteral stent removal (Fig. 3c(i)). We compared the similarity of microbial composition between samples from the same patient (intra-individual) and between different patients (inter-individual) at different sampling

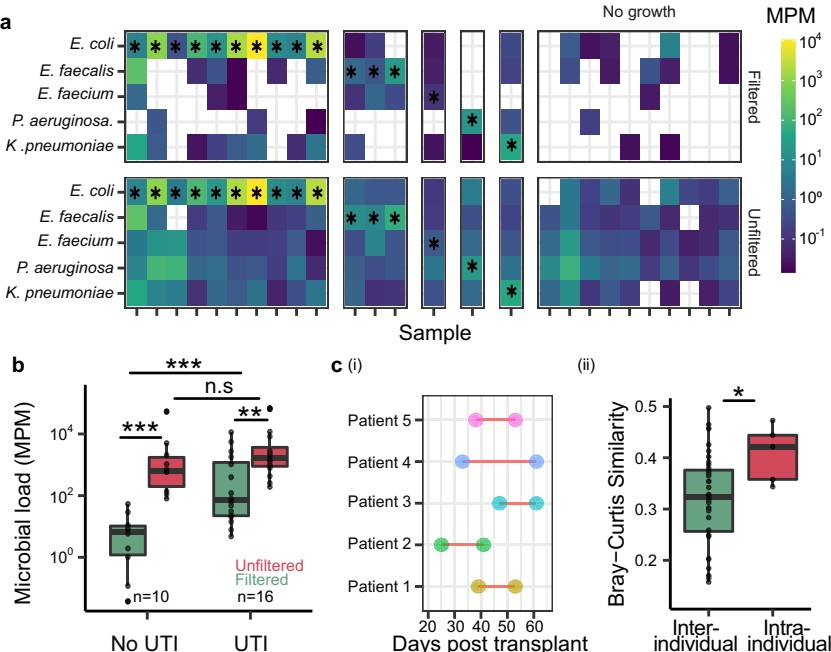

**Fig. 3 Application of SIFT-seq to urine. a** Heatmap of abundance of species (molecules per million, MPM, species with at least one read detected by BLAST) identified in patients with and without urine culture-confirmed UTIs, before and after application of SIFT-seq filter (black * indicates agreement with urine culture). **b** Boxplot of the relative number of microbe-derived molecules (MPM) in samples from patients with and without urine culture-confirmed UTIs, before and after SIFT-seq filtering. **c (i)** Sample collection timepoints after transplantation for 5 patients. **(ii)** Boxplot showing Bray–Curtis similarity index (as defined in **c (i)**) of the urine microbiome within individual patients and between patients before and after stent removal. Boxes in the boxplots indicates 25th and 75th percentile, the band in the box indicated the median and whiskers extend to 1.5 × Interquartile Range (IQR) of the hinge. Outliers (beyond 1.5 × IQR) are plotted individually. (* $p$ value < 0.05, ** $p$ value < 0.01,*** $p$ value < 0.001). Source data for (**a**–**c**(ii)) are provided as a source data file.

points. Using filtered but not the unfiltered datasets, we observed that the microbial composition remained more similar in the same patient (Fig. 3c(ii)) than between different patients, supporting the utility of SIFT-seq to measure subtle dynamics in urine microbiome composition (Mean Bray–Curtis Similarity: $0.41 \pm 0.06$ and $0.317 \pm 0.09$ respectively, $p$ value $= 2.8 \times 10^{-2}$, two tailed, Wilcoxon rank-sum test, Supplementary Fig. 1).

To evaluate the performance of SIFT-seq to existing bioinformatic techniques for eliminating environmental DNA contamination, we benchmarked SIFT-seq against Low Biomass Background Correction (LBBC)[6], a bioinformatics noise filtering tool for eliminating environmental DNA contamination. LBBC identifies and removes two types of noise: (1) digital cross talk stemming from alignment errors and (2) physical noise arising from environmental DNA contamination present in reagents required for DNA isolation and sequencing libraries preparation. We compared SIFT-seq-filtered and LBBC-filtered data for samples from the kidney transplant cohort ($n = 26$). On average, LBBC filtering resulted in a 1.4-fold reduction of reads originating from contaminant genera, while SIFT-seq achieved a 7.5-fold reduction ($p$ value$_{SIFT-seq}$ < 0.001, $p$ value$_{LBBC}$ < 0.001, two-tailed, Wilcoxon signed-rank test) (Supplementary Fig. 2a). SIFT-seq identified all species detected from conventional urine culture (16/16) while LBBC only detected 10/16 species reported by culture (Supplementary Fig. 2b). The decrease in false positive rate after LBBC filtering occurred at the expense of decreased true positive rate. We also performed SIFT-seq on negative controls included in 32/33 experimental batches (see Methods). We quantified the reads originating from contaminant genera before and after SIFT-seq filtering and found that SIFT-seq removed 95.8% of all contaminant genera detected in the negative controls ($506.7 \pm 827.53$ versus $0.4 \pm 0.6$ MPM before and after SIFT-seq filtering, respectively ($p$ value < 0.001, two-tailed, Wilcoxon signed-rank test) (Supplementary Fig. 3).

**SIFT-seq identifies bacterial and viral co-infection of COVID-19 from blood.** The COVID-19 pandemic is an unprecedented human health crisis. Viral or bacterial co-infection occurs in roughly 4% of hospitalized COVID-19 patients but can occur in up to 30% of COVID-19 patients admitted to the intensive care unit[22]. Co-infection has been associated with longer fever duration, and increased risk of intensive care unit admission and need for mechanical ventilation[23]. We reasoned that SIFT-seq may offer sensitive detection of bacterial and viral co-infection in COVID-19 patients with improved specificity over conventional metagenomic sequencing assays.

We applied SIFT-seq to 30 plasma samples from 14 patients with COVID-19 collected as part of a clinical study aimed at identifying predictors of disease severity. Respiratory and blood cultures were obtained as part of standard clinical care. Three patients (P16, P24, P39) tested positive for bloodstream infection and respiratory tract infection, while all other patients were not diagnosed with COVID-19 co-infection. SIFT-seq identified the causative pathogen in 3/3 bloodstream infection cases and 8/8 respiratory infection cases (Fig. 4a, b). Conventional metagenomic sequencing (without SIFT-seq filtering) was equally sensitive to these pathogens but was limited by specificity. Of interest, while we did not obtain plasma collected the day of infection for P24, we identified cfDNA originating from *K. pneumoniae* and *Haemophilus influenzae*, for which the patient tested positive 4 days later. These results suggest that SIFT-seq may be able to identify cases of infection earlier than traditional culture methods, and with improved specificity compared to conventional metagenomic sequencing techniques.

**SIFT-seq identifies infection-causing pathogens in sepsis patients.** Sepsis is a life-threatening organ dysfunction caused by dysregulated host response to a bacterial, viral, fungal or parasitic

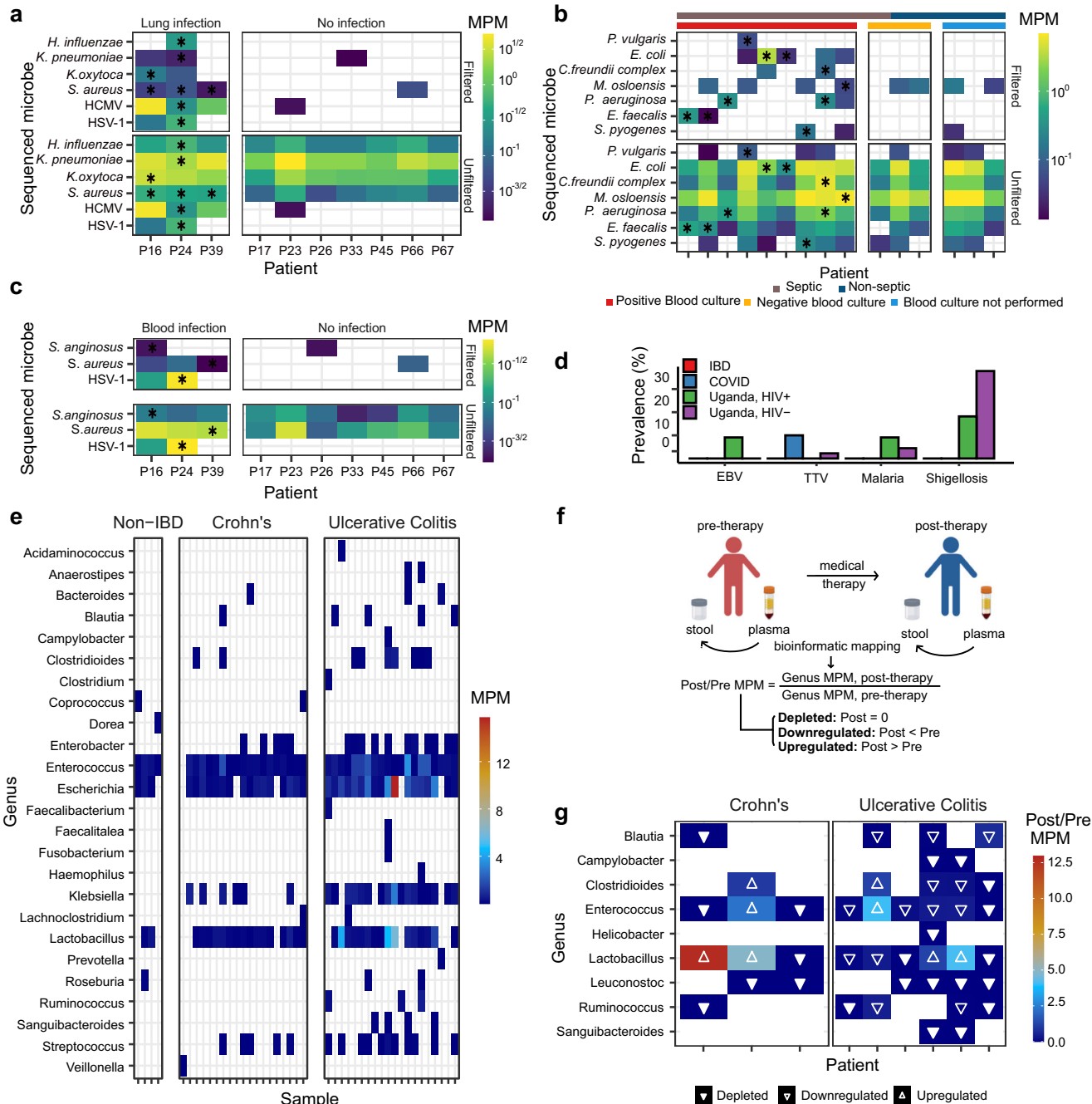

**Fig. 4 Application of SIFT-seq to plasma.** Heatmaps of the abundance of species identified in plasma from COVID-19 patients with and without culture confirmed (**a**) lung and (**b**) blood infection, before and after application of SIFT-seq filter (black * indicates agreement with culture; HCMV: Human cytomegalovirus, HSV-1: Herpes simplex virus 1). **c** A heatmap of abundance of species identified in the sepsis cohort before and after SIFT-seq filtering (black * indicates species identified by blood culture). **d** Barplot of the prevalence of Epstein-Barr Virus (EBV), Torque teno virus (TTV), malaria-causing, or shigellosis-causing microorganisms in different patient cohorts. **e** Heatmap of the abundance of species identified in matched stool and plasma cfDNA samples in patients diagnosed with Crohn's disease or ulcerative colitis. **f** Schematic for matched stool and plasma samples from individuals before and after medical therapy. **g** Heatmap of the change in abundance of gut-specific bacteria before and after treatment. Source data are provided as a source data file.

infection[24]. According to the World Health Organization, in 2017 there were 48.9 million sepsis cases and 11 million sepsis-related deaths worldwide. When sepsis is suspected, broad-spectrum empiric antibiotics are administered, and tests are performed to identify the infection-causing pathogens. Blood culture is the gold standard method to detect infectious pathogens in the bloodstream, however this method is time consuming and limited to few culturable microbes. Though other molecular tests can shorten time to

results when performed directly on blood, the low microbial burden in blood leads to low sensitivity, low negative predictive values, and detection of only a few specific pathogens[25]. Thus, conventional metagenomic cfDNA sequencing holds promise in identifying sepsis-causing pathogens[26,27].

We tested the utility of SIFT-seq to identify sepsis-causing pathogens in patients with sepsis. For this, we performed a blinded analysis on 15 plasma samples ($n = 10$ from septic patients and

$n = 5$ from non-septic patients. 9/15 patients had a positive blood culture result (9/10 patients with sepsis), 3/15 had negative blood culture and for 3/15 patients blood culture was not performed. A total of 10 pathogens were identified in the 9 blood-culture positive samples. After unblinding we found a strong agreement between pathogens that were identified by blood culture and those that were identified by SIFT-seq: SIFT-seq detected 10 out of 10 of pathogens reported by blood culture. Importantly, for only 2/9 patients with positive blood culture, a plasma sample was collected at the time of the positive blood culture (*E. faecalis* identified by blood culture and SIFT-seq, Fig. 4c). For 7/9 patients, the plasma sample for SIFT-seq was collected after the initial positive blood culture and after initiation of antibiotic treatment. Blood cultures corresponding to the time of plasma sample collection for those 7/9 samples were all negative, while SIFT-seq correctly identified the pathogen identified by culture in the sample before initiation of antibiotic treatment. This experiment demonstrates the utility of SIFT-seq to identify blood-borne pathogens in the setting of sepsis, even after initiation of antibiotic treatment when blood cultures frequently fail.

**SIFT-seq identifies clinically-relevant bacterial and viral microorganisms with low prevalence and low microbial burden**. Neglected tropical diseases significantly impact the public health and economies of low-income countries. Treatments exist for many of these diseases, but development and deployment of reliable diagnostic tests has been slow[28]. We reasoned that SIFT-seq could be used to screen for infections with low prevalence and low microbial burden.

We applied SIFT-seq to 56 plasma samples from 44 individuals who presented with symptoms of respiratory illness at outpatient clinics in Uganda. Nine of these individuals were HIV positive at the time of sample collection. We mined the data to determine the prevalence of clinically-relevant bacterial and viral microorganisms endemic to Uganda and compared with results obtained for plasma samples collected from subjects that live in North America (53 plasma samples from the IBD cohort; 30 plasma samples from the COVID-19 cohort). We screened the samples for Epstein-Barr virus, Torque Teno virus, and pathogens associated with malaria (*Plasmodium vivax* and *P. falciparum*), and shigellosis (*Shigella sonnei*, *S. dysenteriae*, *S. boydii*, and *S. flexneri*) before (Supplementary Fig. 4) and after SIFT-seq filtering (Fig. 4d). After SIFT-seq filtering, these microorganisms were found at varying rates in samples from the Uganda cohort: malaria (3/44), Epstein-Barr virus (1/44), shigellosis (19/44), and torque teno virus (1/44), but not in the IBD cohort. Torque teno virus, which has previously been reported to be elevated in immunocompromised patients[8], was identified in 3/30 COVID-19 patient samples, all from patients who had received a bone marrow transplant prior to sample acquisition.

**SIFT-seq identifies signatures of bacterial translocation from the gastrointestinal tract**. Bacterial translocation of intestinal microbes through mucosal membranes is believed to be a normal phenomenon, but has been found to occur more frequently in patients experiencing gut flora disruption[29,30]. In patients with IBD, gut vascular barrier disruption has been linked to increased intestinal permeability and subsequent microbial translocation across the mucosal membrane[31,32]. The translocation of gut bacteria and their products to extraintestinal sites can result in systemic inflammation, resulting in autoimmune or other non-infectious diseases[31,33]. Detecting signatures of translocation is therefore important but difficult in view of the low abundance of microbial DNA due to translocation in blood.

To identify signatures of bacterial translocation, we compared whole genome shotgun sequencing of fecal samples from 44 patients (non-IBD $n = 4$, Crohn's $n = 19$, ulcerative colitis, $n = 21$) to matched plasma cfDNA samples assayed using SIFT-seq. We first quantified bacterial species identified in matched fecal and plasma samples (Fig. 4e). We identified cfDNA derived from gut-specific microbes in all patient samples, though to a much greater extent in individuals with ulcerative colitis ($0.57 \pm 0.65$, $1.22 \pm 1.38$, and $5.55 \pm 9.46$ MPM of gut-specific bacteria for non-IBD, Crohn's disease, and ulcerative colitis samples, respectively). To investigate the effects of treatment on bacterial translocation, we collected additional stool and plasma samples from nine patients (Crohn's $n = 3$, ulcerative colitis $n = 6$) after treatment initiation and performed whole genome shotgun sequencing of stool and SIFT-seq on plasma cfDNA (Fig. 4f). We quantified the relative abundance of gut-specific bacterial species before and after treatment and found that the burden of cfDNA decreased for most bacterial species (28/36) following treatment, which may be explained by a reduction in the degree of bacterial translocation with treatment (Fig. 4g). Of interest, out of seven subjects for which we detected *Lactobacillus* before treatment, five displayed an increase in *Lactobacillus* species burden in blood after treatment (up to 12.7-fold increase after treatment and an average of 3.36-fold MPM increase after treatment across all samples). *Lactobacillus* has been shown to promote gastrointestinal barrier function, protecting the gut from pathogenic bacteria and preventing inflammation[32]. For bacterial species besides *Lactobacillus*, we find an average of 70% reduction in MPM after treatment. These preliminary results support the use of SIFT-seq to identify subtle signatures of bacterial translocation in the blood.

## Discussion

We report SIFT-seq, a method for metagenomic DNA sequencing that is robust against DNA contamination. In contrast to prior methods for the management of DNA contamination that have relied on algorithmic batch correction or the use of known-template or no-template controls, SIFT-seq uses a physical labeling technique to differentiate sample-intrinsic DNA from contaminating DNA. The principle of SIFT-seq has the potential for broad application in contexts where metagenomic analyses of isolates with low biomass of microbial DNA are required. In this proof-of-principle study, we have explored applications of SIFT-seq to quantify microbial cfDNA in human biofluids. Metagenomic sequencing of microbial cfDNA in blood or urine is a highly sensitive approach to screen for a broad range of viral or bacterial pathogens, but because of the low biomass of microbial DNA in blood and urine this method is highly susceptible to DNA contamination leading to a high false positive rate. We implemented SIFT-seq tagging of cfDNA in plasma and urine by bisulfite-induced deamination of unmethylated cytosines and show that this approach reduces background signals from common contaminants by up to three orders of magnitude. SIFT-seq thereby dramatically improves the specificity of metagenomic cfDNA analyses, opening up a broad range of applications, e.g., infectious disease with low microbial burden or syndromes that are accompanied by subtle changes in the plasma or urine microbiome.

In its current implementation, SIFT-seq has several limitations. First, SIFT-seq is only robust against DNA contamination introduced after the labeling step. We implemented SIFT-seq tagging directly on biofluids, which allowed us to identify contaminants introduced during DNA isolation or library preparation but not during the sample collection or isolation of the plasma from whole blood. In addition, SIFT-seq does not account for

cross-contamination or index hopping during DNA isolation and library preparation. Thus, other complementary best practices, such as the incorporation of unique molecular identifiers and non-redundant dual indices, for reducing contamination in DNA studies should be adopted as necessary[4]. Second, the specific labeling strategy we have implemented here inherently modifies the DNA sequence and thereby limits the resolution of SNP or other per-base analyses but does not impact the detection of microbial cfDNA and provides opportunities to evaluate the host response to injury[12]. Alternative implementations of contamination-free sequencing that do not introduce sequence alterations can be considered. Last, the principles introduced here can be adopted for molecular assays beyond whole genome sequencing, including amplicon sequencing assays, e.g., 16 S rRNA profiling, or PCR assays.

## Methods

### Study cohort and sample collection

*Uganda cohort and sample collection.* Forty-four plasma samples were collected from individuals that presented with respiratory symptoms at outpatient clinics in Uganda. Briefly, peripheral blood was collected in Streck Cell-Free BCT (Streck #230257) and centrifuged at $1600 \times g$ for 10 min. Plasma was stored in 1 mL aliquots at −80 °C. The study was approved by the Makerere School of Medicine Research and Ethics Committee (protocol 2017-020). All patients provided written informed consent.

*IBD cohort sample collection.* Peripheral blood samples were collected under IRB approved protocol (1806019340) at the Jill Roberts Center for IBD at Weill Cornell Medicine. PBMCs and plasma were fractionated using a Ficoll-Hypaque gradient. Informed consent was obtained from all participants.

*IBD cohort fecal sample collection.* DNA from fecal samples was isolated using the MagAttract PowerMicrobiome DNA/RNA kit with glass beads (Qiagen, Germany). Metagenomic libraries were prepared using the NEBNext Ultra II for DNA Library Prep kit (New England Biolabs, Ipswich, MA) following the manufacturer's protocol. The DNA library was sequenced on an Illumina HiSeq instrument using a $2 \times 150$ paired-end configuration in a high output run mode. Informed consent was obtained from all participants.

*COVID-19 cohort sample collection.* Peripheral blood samples were collected as part of an observational study among individuals with COVID-19 that were treated at NewYork-Presbyterian/Weill Cornell Medical Center (NYP-WCMC) and Lower Manhattan Hospital under IRB approved protocol (IRB 20-03021645). Informed consent was obtained from all participants. PBMCs and plasma were fractionated using a Ficoll-Hypaque gradient.

*Sepsis cohort sample collection.* Since 2014, investigators have prospectively consented patients admitted to any ICU at NYP-WCMC to participate in a registry involving collection of biospecimens and clinical data. For each participant, whole blood (6–10 mL) was obtained. whole blood samples were drawn into EDTA-coated blood collection tubes (BD Pharmingen, San Jose, CA). Samples were stored at 4 °C and centrifuged within 4 h of collection. Plasma was separated and divided into aliquots and kept at −80 °C.The registry was approved by the institutional review board of WCMC (1405015116, 20-05022072).

*Kidney transplant cohort sample collection.* Twenty six urine samples were collected from 23 kidney transplant recipients who received care at NYP-WCMC. The study was approved by the Weill Cornell Medicine Institutional Review Board (protocol 1207012730). All patients provided written informed consent. Patients provided urine specimens using a clean-catch midstream collection protocol. The urine specimen was centrifuged at $3000 \times g$ for 30 min and supernatant was stored as 1 mL of 4 mL aliquots.

*Early post transplant sample collection.* Urine specimens collected within $10 \pm 5$ days of ureteral stent removal from patients who agreed to participate in the WCM IRB approved protocol # 20-01021269 were included in this study. Urine specimens were collected within $47 \pm 11$ days post-kidney transplantation. The presence of UTI was excluded by a negative urine culture and the absence of pyuria. This study was approved by the Weill Cornell Medicine Institutional Review Board (protocol 20-01021269).

### Definition of positive and negative urine culture for the UTI and early post-transplant cohorts
A positive urine culture was defined as a culture growing an organism identified to at least the genus level (≥10,000 cfu/mL). A urine culture was defined as negative when either no organism was isolated in culture

(<1000 cfu/mL) or the organism was unidentified to either the genus or species level (i.e., unidentified) and the colony count was <10,000 cfu/mL.

### SIFT-seq in plasma
An aliquot of 520 μL of plasma was centrifuged at $20,000 \times g$ (~14,000 RPM) for 10 min at 12 °C to pellet cellular debris. The supernatant was transferred to a new 1.5 mL tube and the final volume was brought up to 1000 μL with PBS. The solution was heated to 98 °C for 10 min and mixed at $190 \times g$(1000 RPM) to coagulate the albumin present in plasma. The solution was then centrifuged at $1600 \times g$ (~4000 RPM) for 10 min. 500 μL of supernatant was transferred to 15 mL falcon tube containing 3.25 mL of ammonium bisulfite solution (Zymo Research, product #5030) and shaken in a thermomixer at 98 °C for 10 min (15 s on/30 s off). Samples were then transferred to a thermomixer at 54 °C for 60 min (15 s on/30 s off). Then, cfDNA extraction was performed using the QIAamp Circulating Nucleic Acid Kit using the 4 mL plasma protocol (Qiagen, product #55114). Prior to DNA elution, 200 μL of L-Desulphonation buffer (Zymo Research, product #5030) was added to the columns for 15 min, followed by two washes with 200 μL absolute ethanol. DNA was then eluted according to manufacturer recommendations, and single-stranded library preparation is performed (Claret Biosciences, product #CBS-K150B). Libraries were then sequenced on an Illumina sequencer. Step by step protocol is provided in the supplementary information file.

### SIFT-seq in urine
An aliquot of 520 μL of urine was centrifuged at $20,000 \times g$(~14,000 RPM) for 5 min to pellet cellular debris. 500 μL of supernatant was transferred to a new 15 mL falcon tube containing 3.25 mL of ammonium bisulfite solution (Zymo Research, product #5030) and heated to 98 °C for 10 min. Samples were then kept at 54 °C for 60 min. Then, cfDNA extraction was performed using a commercially available column-based kit (Norgen Biotek, product #56700). Prior to DNA elution, 200 μL of L-Desulphonation buffer (Zymo Research, product #5030) was added to the columns for 20 min, followed by two washes with 200 μL absolute ethanol. DNA was then eluted according to manufacturer recommendations, and single-stranded library preparation was performed (Claret Biosciences, product #CBS-K150B). Libraries were then sequenced on an Illumina sequencer. Step by step protocol is provided in the supplementary information file.

### Sequencing library preparation
Bisulfite conversion of cfDNA involves a cfDNA denaturing step at 98 °C such that we get single stranded cfDNA molecules after DNA extraction. For this reason, a single stranded sequencing library preparation method is chosen for the next steps. We prepared sequencing libraries using the SRSLY PicoPlus DNA NGS Library Preparation Base Kit (SRSLY Cat# CBS-K250B-24) with the SRSLY UDI Primer Set-24 (SRSLY Cat# CBS-UD-24) following the manufacturer's protocol, with the following modifications:

1. The input cfDNA volume used was 18 μL.
2. 1.25 μL of NGS Adapters A and 1.25 μL of NGS Adapters B were added to the 20 μL denatured DNA reaction tube, and the volume was completed by 1.5 μL of ultrapure water.
3. The Index PCR Master Mix was substituted for an equal volume of KAPA HiFi Uracil+ Ready Mix (2×).
4. The Indexed Library DNA Purification step was performed twice, first eluting in 50 μL and then in 25 μL.

### Alignment to the human genome
Adapter and low-quality bases from the reads were trimmed using BBDuk (BBDuk V38.46[34],–entropy = '0.25'–maq = '10' -Xmx1g tbo tpe) and aligned to the C-to-T and G-to-A converted human genome using Bismark (Bismark-0.22.1[35],–unmapped,–quiet). PCR duplicates were removed using Bismark.

### Depth of coverage
The depth of sequencing was measured by summing the depth of coverage for each mapped base pair on the human genome after duplicate removal, and dividing by the total length of the human genome (hg19, without unknown bases).

### Removing unconverted molecules
Aligned BAM files are filtered to remove unconverted molecules using the Bismark (Bismark-0.22.1) alignment package with default parameters.

### Bisulfite conversion efficiency
We estimated bisulfite conversion efficiency by quantifying the rate of C[A/T/C] methylation in human-aligned reads (using MethPipe V3.4.3[36]), which are rarely methylated in mammalian genomes.

### Pre-processing of the unmapped reads
Reads originating from the Phix genome were removed from the host unmapped reads using Bowtie 2[37] (Bowtie 2.4.3,–local,–very-sensitive-local,–un-conc). Read IDs from the remaining reads were used to subset paired-end reads from the original FASTQ files. Adapter trimming and read quality filtering was performed using BBDuk (BBDuk

V38.46, maq = 32). Remaining reads were deduplicated using samtools[38] (samtools V1.14) and merged using FLASH2[39] (-q -M75 -O). K-mer decontamination to remove human reads was then performed using BBDuk (BBDuk V38.46, k = 50, prealloc = t) and the obtained fastq file was converted to a fasta file for metagenomics analysis.

**Metagenomic abundance estimation from sequencing data.** Reads mapping to microbial species were identified using HS-BLASTN[40] (hs-blastn-1.0.0) and microbial abundances were estimated using GRAMMy (version 1)[41]. Specific to SIFT-seq, read-level filtering of contaminants is performed by removing sequenced reads with 4 or more cytosines present, or one methylated CpG dinucleotide (the latter represents unmapped, human-derived molecules). Species-level filtering based on the distribution of mapped reads is carried out by first aligning filtered and unfiltered datasets independently. Cytosine-densities of mapping-coordinates in both datasets are measured using custom scripts, and their distributions are compared using a Kolmogorov–Smirnov test. Significantly different filtered-unfiltered distributions are further processed (D-statistic > 0.1 and p value < 0.01). Briefly, filtered datasets whose distribution of cytosines at mapped locations is significantly lower than unfiltered datasets have one read removed, and are re-tested for differences in their distribution. If the distributions are more similar (as measured through the same criteria), it is filtered out. This process is repeated until distributions are no longer significantly different, or if all reads are removed. Read and species level filtering was performed using custom scripts written in Python. Microbial abundance in downstream analyses was quantified as Molecules Per Million reads (MPM) [Eq. 1].

$$MPM = \frac{Adjusted\ Blast\ hits \times 10^6}{Total\ Trimmed\ Reads} \quad (1)$$

**Identification of translocated gut bacteria in plasma.** Fecal shotgun metagenomic data for 53 samples was obtained from 44 patients diagnosed with IBD. Low-quality bases and Nextera-specific sequences were trimmed using Trim Galore V 0.6.5 (Trim Galore V 0.6.5,–nextera–paired). Reads were aligned against the human references (UCSC hg19) using Bowtie2[37] (Bowtie 2.4.3,–maxins 700–no-discordant–score-min L,0,–0.2). Unaligned reads were extracted and assembled with metaSPAdes[42] (SPAdes 3.15.3;–meta) and classified with Kaiju (Kaiju 1.7.4)[43]. Paired cfDNA samples were filtered with SIFT-seq pipeline and aligned to the assembled reads with Bismark (Bismark 0.22.1). Mapped reads with a minimum quality score of 15 were extracted and filtered for gut-specific microorganisms identified by The Human Gut Microbiome Atlas[44].

**Benchmarking SIFT-seq against low background biomass correction (LBBC).** To benchmark SIFT-seq we compared its performance to Low Microbial Biomass Background Correction (LBBC) tool. For this we used datasets from the UTI cohort from our study and matched standard sequenced datasets from a previously published study[11]. Default LBBC filtering parameters were used for this analysis ($\Delta CV_{max} = 2$, $\delta^2_{min} = -5.5$).

**Statistics and reproducibility.** All statistical methods were performed in R version 4.0.5. Groups were compared using a two-sided Wilcoxon Signed Rank or Wilcoxon Rank-Sum tests. Boxes in the boxplots indicates 25th and 75th percentile, the band in the box indicated the median and whiskers extend to $1.5 \times$ Interquartile Range (IQR) of the hinge.

We collected as many patient samples as possible that fit the criteria in each cohort. A detailed description of the sample sizes in each cohort is given in Supplementary Table 1. We excluded data from three samples that were collected by Foley from the Kidney Transplant cohort samples, included samples were all collected by clean-catch method; we also excluded data from four samples that had mixed urine culture results or associated with positive urine culture from the Early Post Transplant cohort, included samples were all urine culture negative. Investigators were blinded to group allocation during data collection of samples in the Sepsis cohort. Groups, and detailed clinical information (e.g., data from conventional blood cultures) were shared with the investigators after the data was analyzed and shared with collaborators who then shared metadata elements. For the other groups, blinding was not implemented the study was focused on the development of a new method and because in the case of the Kidney Transplant and uganda cohorts group allocations were available from prior studies by the same investigators. Experiments were not randomized.

**Reporting summary.** Further information on research design is available in the Nature Research Reporting Summary linked to this article.

## Data availability
ΦX174 DNA sequencing data used in the proof-of-principle experiments has been deposited in NCBI's Sequence Read Archive (SRA) under Bioproject ID PRJNA782310. Sequencing data from human plasma and urine cfDNA is available in the database of Genotypes and Phenotypes (dbGaP), accession number phs001564.v1.p1. Source data are provided with this paper.

## Code availability
All scripts used in this study are available at https://github.com/omrmzv/SIFTseq (https://doi.org/10.5281/zenodo.6622189)[45].

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

## Acknowledgements

We thank the Cornell Genomics Center for help with sequencing assays, the Cornell Bioinformatics facility for computational assistance and Lars Westblade for helpful discussions. A special thanks to Dr. Alfred Andama for his research supervision with the Infectious Disease Research Collaboration (IDRC) for samples collected and character-ized in Kampala, Uganda. We are also grateful to Dr. Inghirami and the members of the Immunopathology laboratory at New York Presbyterian Hospital, Weill Cornell Medi-cine, for the support in covid-19 sample processing. This work was supported by R01AI146165 (to I.D.V.), R21AI133331 (to I.D.V.), R21AI124237 (to I.D.V.), DP2AI138242 (to I.D.V.), R01AI151059 (to I.D.V., J.R.L., M.Su., C.E.M., D.D.), R37 AI051652 (to M.Su.), NIH U01DA053941 and STARR I13-0052 (C.E.M.) a Synergy award from the Rainin Foundation (to I.D.V. and R.L.), NHLBI K23 HL151876 (to E.J.S), a Cornell University's Ignite Acceleration grant, a grant from the Bill and Melinda Gates Foundation INV-003145 (to I.D.V.). A.C. was supported by the National Institutes of Health under the Ruth L. Kirschstein National Research Service Award (6T32GM008267) from the National Institute of General Medical Sciences. A.P.C. is supported by a National Sciences and Engineering Research Council of Canada PGS-D3 fellowship. Figures 1(a) and 4(f) were created with BioRender.com.

## Author contributions

O.M., A.P.C., A.C., and I. D.V. conceived and designed the study. O.M., A.P.C., A.C., J.S.L., and L.K.D. performed the experiments. R.L., A.S., L.G.G., E.J.S., M.Sa., M.J.S., M.Su., J.R.L., C.E.M., D.D. identified and collected patient samples and clinical metadata. O.M., A.P.C., A.C., S.S. and I.D.V. analyzed the data. R.L., A.S., E.J.S., M.Sa., M.J.S., M.Su., J.R.L., C.E.M., D.D. and I.D.V. aided in interpretation of the results. O.M., A.P.C., A.C., and I.D.V wrote the paper. All authors provided comments and edits. O.M., A.P.C., and A.C. contributed equally.

## Competing interests

I.D.V., O.M., A.P.C., and A.C. have submitted a patent related to the present work. A.P.C., I.D.V., D.D., and J.R.L. are inventors on the patent US-2020-0048713-A1 titled "Methods of Detecting Cell-Free DNA in Biological Samples." I.D.V. is a member of the Scientific Advisory Board of Karius Inc., Kanvas Biosciences and GenDX. J.R.L. received research support under an investigator-initiated research grant from BioFire Diagnostics, LLC. E.J.S. is a consultant for Axle Informatics. Remaining authors declare no competing interests.
