## [Peer Review File · Nature Communications]

REVIEWERS' COMMENTS

Reviewer #1 (Remarks to the Author):

The reviews have appropriately addressed my comments, as reviewer #1, on the first iteration of this manuscript leading to a significantly improved article.

Reviewer #2 (Remarks to the Author):

Thank you for providing me with a copy of the updated manuscript, "A metagenomic DNA sequencing assay that is robust against environmental DNA contamination". Most of the points raised by myself and other reviewers have been addressed in the revisions or discussed in the rebuttal.

In particular I feel that the paper has been strengthened by the addition of a comparison with LBBC and the blinded sepsis analysis.

Although residual contaminant reads remain after SIFT-seq processing, the heatmaps and negative control data do show this patently, so the reader can draw their own conclusion about suitable further steps to account for background. The discussion makes it clear that SIFT-seq is an additional tool for a decontamination toolkit and not a complete solution, which was a notable concern for all reviewers.

I would be happy to recommend the revised manuscript for publication.

Reviewer #3 (Remarks to the Author):

authors have addressed my feedback

Point-by-point address of the reviewer comments.

(Our response in blue font)

We thank the reviewers for their comments and suggestions that have allowed us to improve our paper.

Reviewer #1 (Remarks to the Author):

The reviews have appropriately addressed my comments, as reviewer #1, on the first iteration of this manuscript leading to a significantly improved article.

Reviewer #2 (Remarks to the Author):

Thank you for providing me with a copy of the updated manuscript, “A metagenomic DNA sequencing assay that is robust against environmental DNA contamination”. Most of the points raised by myself and other reviewers have been addressed in the revisions or discussed in the rebuttal.

In particular I feel that the paper has been strengthened by the addition of a comparison with LBBC and the blinded sepsis analysis.

Although residual contaminant reads remain after SIFT-seq processing, the heatmaps and negative control data do show this patently, so the reader can draw their own conclusion about suitable further steps to account for background. The discussion makes it clear that SIFT-seq is an additional tool for a decontamination toolkit and not a complete solution, which was a notable concern for all reviewers.

I would be happy to recommend the revised manuscript for publication.

Reviewer #3 (Remarks to the Author):

authors have addressed my feedback